# Neuroangiostrongyliasis: Updated Provisional Guidelines for Diagnosis and Case Definitions

**DOI:** 10.3390/pathogens12040624

**Published:** 2023-04-20

**Authors:** Carlos Graeff-Teixeira, Kittisak Sawanyawisuth, Shan Lv, William Sears, Zhaily González Rodríguez, Hilda Hernández Álvarez, Pedro Casanova Arias, Leticia Karolini Walger Schultz, Alicia Rojas, John Jacob, Susan Jarvi, Kenton Kramer

**Affiliations:** 1Center for Health Sciences, Universidade Federal do Espírito Santo, Vitória 29075-910, ES, Brazil; leticia.walgers@gmail.com; 2Department of Medicine, Faculty of Medicine, Khon Kaen University, Khon Kaen 40000, Thailand; yong2001th@yahoo.com; 3Chinese Center for Disease Control and Prevention, National Institute of Parasitic Diseases, Shanghai 200025, China; lvshan@nipd.chinacdc.cn; 4Laboratory of Parasitic Diseases, National Institutes of Health, Bethesda, MD 20892, USA; william.sears@nih.gov; 5Reference National Parasitology Laboratory, Instituto de Medicina Tropical “Pedro Kourí” (IPK), Havana 11400, Cuba; zhaily@ipk.sld.cu (Z.G.R.); hilda@ipk.sld.cu (H.H.Á.); rcapedro@ipk.sld.cu (P.C.A.); 6Departamento de Parasitologia, Facultad de Microbiologia, Universidad de Costa Rica, San Jose 11501-2060, Costa Rica; 7Department of Pharmaceutical Sciences, Daniel K. Inouye College of Pharmacy, University of Hawaii at Hilo, Hilo, HI 96720, USA; 8Department of Tropical Medicine, Medical Microbiology and Pharmacology, John A Burns School of Medicine, University of Hawaii at Manoa, Honolulu, HI 96822, USA

**Keywords:** angiostrongyliasis, neuroangiostrongyliasis, rat lungworm disease, eosinophilic meningitis

## Abstract

*Angiostrongylus cantonensis* is the main causative agent for eosinophilic meningoencephalitis in humans. Larvae are rarely found in the cerebral spinal fluid (CSF). Consequently, serology and DNA detection represent important diagnostic tools. However, interpretation of the results obtained from these tools requires that more extensive accuracy studies be conducted. The aim of the present study is to update guidelines for diagnosis and case definitions of neuroangiostrongyliasis (NA) as provided by a working group of a recently established International Network on Angiostrongyliasis. A literature review, a discussion regarding criteria and diagnostic categories, recommendations issued by health authorities in China and an expert panel in Hawaii (USA), and the experience of Thailand were considered. Classification of NA cases and corresponding criteria are proposed as follows: minor (exposure history, positive serology, and blood eosinophilia); major (headache or other neurological signs or symptoms, CSF eosinophilia); and confirmatory (parasite detection in tissues, ocular chambers, or CSF, or DNA detection by PCR and sequencing). In addition, diagnostic categories or suspected, probable, and confirmatory are proposed. Updated guidelines should improve clinical study design, epidemiological surveillance, and the proper characterization of biological samples. Moreover, the latter will further facilitate accuracy studies of diagnostic tools for NA to provide better detection and treatment.

## 1. Introduction

*Angiostrongylus cantonensis* is an intra-arterial nematode which, in accidental human hosts, can cause eosinophilic meningitis [1]. Eosinophilic inflammatory responses in the central nervous system (CNS) are mainly elicited by helminths. However, these inflammatory responses can also occur in association with cancer, intra-thecal drugs, and intra-vesicular devices [2]. The non-infectious causes for eosinophilic meningoencephalitis are dominant in many regions, e.g., Central Europe.

*A. cantonensis* are parasites that live inside the pulmonary arteries of rodents, and mollusks serve as intermediate hosts. Several other invertebrates, such as shrimp, frogs, and lizards, can serve as paratenic hosts. Larvae of *A. cantonensis* develop in the fibromuscular tissues of mollusks, and subsequently infect humans when raw or undercooked food is ingested [1,3]. Larvae-contaminated water may also be a source of human infection [4,5]. In accidental hosts, including humans, larvae migrate and are retained in the CNS. This retention prevents complete maturation of adult worms inside the pulmonary arteries [1,3].

Larvae are rarely detected in examinations of cerebrospinal fluid (CSF), thereby preventing confirmation of a diagnosis by direct identification of parasites [6]. Therefore, immunological and DNA detection methods are important tools for evaluating patients with suspected neuroangiostrongyliasis (NA) [7,8,9]. While antibody and DNA-detection methods have been standardized, evaluations of these methods as reliable detection tools have been less than adequate due to the small number of well-characterized biological reference samples that are currently available from different geographical areas. More recently, a highly sensitive and specific quantitative PCR method has been developed to confirm diagnosis of NA [10].

*A. cantonensis* is native to southeastern Asia and the Pacific Islands [11], yet its presence has expanded to multiple continents. This perceived expansion may also have resulted from an increased awareness and availability of diagnostic capability. It is also a food-borne disease that has been linked to travelers. A rough estimate of the cumulative number of reported cases worldwide is 2800 [3]. Despite the potential for severe CNS disease, NA is not a highly prevalent infection worldwide. To date, the Hawaiian Islands and southeastern Asia, especially Thailand, have been the most affected endemic areas. However, reduced rates of occurrences have been observed in recent years [12,13].

Disease caused by *A. cantonensis* includes isolated meningeal lesions and meningitis associated with brain tissue inflammation (meningoencephalitis) [14]. More rarely, ocular angiostrongyliasis can develop [15]. For severe cases of encephalitis, lethality may reach 80% [16]. The objective of the present work is to update and explore a possible consensus regarding diagnostic criteria and case definitions for NA. It is anticipated that such effort will improve patient management worldwide, will promote comparable clinical and epidemiological studies, and will define conditions for establishing an international biobank of well characterized samples. The latter would represent a valuable resource for evaluations of diagnostic tests to achieve detection and control of NA.

## 2. Materials and Methods

Several meetings of the International Symposium on Angiostrongyliasis (ISA), also named the International Symposium on Rat lungworm Disease, have been held in various countries over many years, namely, Thailand, China, Hawaii (USA), and Australia. The Canary Islands will host the next ISA in September 2023. When a group of researchers and clinician attendees of the ISA met in an online discussion in October 2021, the need for more extensive studies, including multicenter accuracy studies, of diagnostic tests for NA was highlighted. Basic requirements for such studies are: (i) clear and well-defined diagnostic criteria; and (ii) establishment of a collaborative international biobank with well-characterized biological samples. On 23 November 2021, the International Network on Angiostrongyliasis was established. Subsequently, several online meetings were hosted throughout 2022 to discuss many issues, including diagnostic criteria. Health authorities in China and an expert panel in Hawaii (USA) have independently established recommendations for diagnosing and treating NA [17,18]. Diagnostic and patient management experience from Thailand, currently the most endemic country, are also available. In particular, the results from a systematic review published by Khamsai and collaborators (2020) were examined [13]. Based on these considerations, the following revised guidelines were developed. They are not to be definitive but represent a starting point for continuous improvement.

## 3. Results

The present updated guidelines were developed based on the following principles: (i) to provide clear definition and expression (“taxonomy”) of symptoms, signs, or laboratory results; (ii) to select the most relevant symptoms most closely related to NA (sensitivity) and classify them as minor, major, or confirmatory criteria (meanwhile, excessive value for general, less specific symptoms and signs is avoided); and (iii) to define categories of diagnosis according to the degree of certainty for etiological diagnosis, from lower (suspected) to highest (probable and confirmed) degrees, conforming to standard organization of diagnostic and treatment guidelines [19,20,21,22,23].

Elements of exposure history are summarized in Table 1. Criteria are classified as minor, major, or confirmatory (Table 2 and Figure 1). Diagnosis categories (suspected, probable, and confirmed) and recommended actions are also presented in Table 3 and Figure 1. Antibody detection (serology), blood eosinophilia, and history of exposure are considered minor criteria, since their isolated presence does not constitute strong evidence for NA. In addition, these criteria may be absent in patients. While serological studies are useful for epidemiological exposure studies [5,24], cross-reactivity and persistence of antibodies after cure are recognized as universal limitations of serology for the detection of current infections. 

Major criteria, headache, or other neurological signs or symptoms associated with CSF eosinophilia are proposed as defining criteria for eosinophilic meningitis or meningoencephalitis. Importantly, these major criteria are labeled as such to be highly suggestive for NA, since *A. cantonensis* is their main causative agent. Clinical manifestations’ strength of evidence for NA depends on the absence of other obvious causes for eosinophilic meningitis or meningoencephalitis [18].

Finding parasitic structures in cerebrospinal fluid or brain tissues is extremely rare yet represents undisputed criteria for confirming a diagnosis. Ocular examination, including anterior and posterior chambers, may also disclose larvae or even adult worms since the eye is an area of localization second to meningeal vessels for *A. cantonensis* [15]. Worms may rarely be found inside pulmonary arteries in fatal cases [27,28,29].

## 4. Discussion

In 2006, the Chinese Ministry of Health published recommendations on diagnostic criteria for angiostrongyliasis and proposed three case definitions: (i) suspected; (ii) clinical, and (iii) parasitologically diagnosed [17]. Eating history (C1), clinical manifestations (C2), blood eosinophilia (C3), CSF eosinophilia (C4), seropositivity (C5), and presence of parasites in CSF or other sites (C6) were the suggested criteria. Suspected NA is considered whenever a combination of two criteria of C1 to C5 is observed. A clinical diagnosis is considered when C1, C2, C3, and C4 are present. NA is confirmed with demonstration of parasites (C6).

In 2018, a State Task Force in Hawaii (USA) prepared guidelines for diagnosis and treatment of NA. An update was subsequently published in 2020 [18]. The latter guidelines consider two diagnostic categories: presumptive and definitive, or confirmed by identification of parasites or DNA detection with PCR. A presumptive diagnosis is established by: (i) characteristic symptoms and signs, (ii) exposure history, and (iii) CSF eosinophilia.

Headache is the most common clinical manifestation and can occur in the absence of other neurological signs or symptoms. Neck stiffness and fever are also present in 15% and 40% of cases, respectively, and occur more often in children [26,30]. Among less common manifestations, dysesthesias (paresthesias and hyperesthesias) and migratory myalgia may be valuable to indicate neuroangiostrongyliasis, which are to be properly investigated in prospective clinical studies [3,6]. Other early symptoms can be prodromal symptoms; that is, those due to the physiological and neurological damages due to the larval migration from the gastrointestinal tract into the CNS [31,32]. It is likely that the severity of prodromal symptoms is directly associated with the number of parasites involved in the infection. Larvae attempting to penetrate the gastrointestinal walls may cause symptoms such as nausea, vomiting, abdominal pain, or diarrhea. Larval migration through the liver, kidneys, and lungs may cause malaise, low-grade fever, coughs, jaundice-like symptoms, and hematuria. Larvae stranded beneath the skin may produce rashes or pruritis-like symptoms. Prodromal symptoms are nonspecific; thus, unless there is a great degree of suspicion of infection, it is unlikely to alert the medical practitioner [32,33].

CSF eosinophilia is another main indicator for NA and occurs in approximately 50% of patients [26]. To date, the definition of “CSF eosinophilia” remains controversial. Some authors consider any number of eosinophils as abnormal, while other authors have selected 10% or an absolute number of 10 eosinophils as a threshold [18,26]. The degree of eosinophilia present may be a stronger indicator for NA, and patients with ≥40% are more likely to have NA [34]. In some cases, only a follow-up lumbar puncture can reveal CSF eosinophilia [3]. However, lack of appropriate staining and differential counting of CSF cells can prevent demonstration of an eosinophilic inflammatory response in meningeal tissues and fluids. Physicians are urged to check with their laboratory to determine if a proper examination was performed.

Different transmission areas may present predominant intentional or nonintentional exposure behavior. For example, in Thailand, intentional food habits favor transmission; while in Hawaii, ingestion of contaminated food or water is usually non-intentional. More severe cases in Hawaii may be due to higher intake of larvae because of the high burden of infection of terrestrial gastropods. A definite exposure history may be absent [18]. The precise date of exposure is important to consider since the incubation period (IP) for NA is usually between 1 and 3 weeks, although a 1-year IP has been reported [3,16,32]. In addition to the well-known role of several foods as a source for infection, there are indications that larvae can be ingested in drinking water (Table 1) [4]. Thus, knowledge of active transmission areas may help increase and sustain awareness regarding NA, help identify cases for early treatment, and promote the prevention of more severe disease. In endemic areas, knowledge and attention to non-specific clinical manifestations (e.g., fever, nausea, vomiting, agitation, lethargy) can facilitate early diagnosis and treatment. There is need for continued research to decrease the time to diagnosis. Perhaps antigen capture assays for blood, stool, or urine will be helpful. Post-exposure treatment with pyrantel has been shown to be promising, although its clinical relevance is yet to be demonstrated, especially considering the very short time frame from exposure to the effective prevention of larvae migration through intestinal mucosa [35].

While several serological tests have been developed, limitations involving cross-reactivity [36,37], late seroconversion [38], and less than adequate accuracy evaluations have prevented these results from being confirmatory. Correspondingly, in the present classification proposed, positive serological examination is recognized as a minor criterion. Seroconversion will be helpful to confirm the diagnosis when CSF cannot be collected and an accurate diagnostic test is available.

The proposed guidelines advocate for early treatment of “probable” cases (without other obvious causes), as well as confirmed NA cases (Table 2). A possible exception for early treatment recommendation is prompt recovery (less than 24 h) from headache and other neurological deficits without any clinical manifestations suggestive of encephalitic compromise. Prospective studies are needed to confirm and optimize diagnostic workflow and case definitions. Corticosteroids are a cornerstone of NA management since they can potentially reduce both the intensity and duration of headaches, the main cause of distress in patients [39,40,41]. Non-steroid anti-inflammatory drugs should not be administered along with corticosteroids because of increased risk for upper gastrointestinal bleeding [39]. Measures to reduce intracranial pressure, such as therapeutic repeated CSF removal, have been shown to be effective, and are a choice symptomatic treatment for alleviating severe headaches in patients affected by NA [39]. Benzimidazole anthelmintic drugs, especially albendazole, are also recommended despite multiple controversies regarding their safety and efficacy [42]. For detailed discussion and recommendations for angiostrongyliasis treatment, see specific reports and reviews [14,25,41].

## 5. Conclusions

In conclusion, it is anticipated that the present global revision and updated recommendations for diagnosis and treatment of NA will facilitate much needed clinical studies, with the use of standardized diagnostic criteria leading to better comparative studies of patients from different geographic areas. The present revisions and updates, intended to support further discussions and developments, may also help provide well-defined diagnostic categories for public health surveillance and a strategy for better characterizing biological samples for accuracy studies of diagnostic tools. The latter is especially relevant for evaluations of newly developed methods for early and specific detection and treatment of NA.

## Figures and Tables

**Figure 1 pathogens-12-00624-f001:**
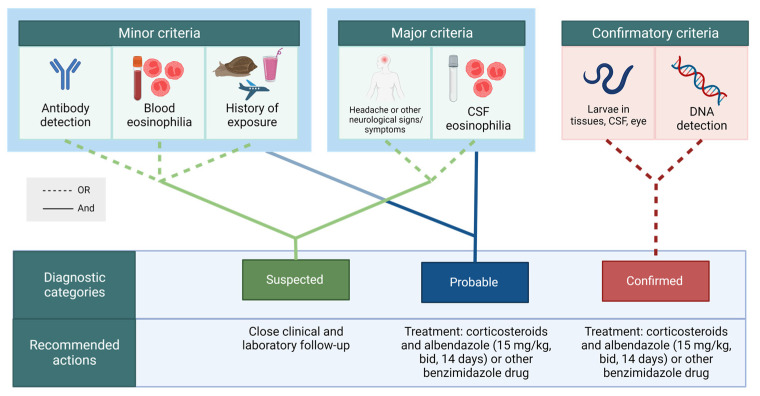
Diagnostic criteria classification, case definitions, and recommendations for follow up and treatment of patients with neuroangiostrongyliasis, as proposed by a working group from the International Network on Angiostrongyliasis.

**Table 1 pathogens-12-00624-t001:** *Angiostrongylus cantonensis* and elements from exposure history with increased risk of transmission, according to Wang et al. [3], Khamsai et al. [25], Ansdell et al. [18], and Howe et al. [4].

Type of Exposure	Vectors/ Transmission Areas
Ingestion of raw, undercooked and/or inadequately washed foods	Mollusks: snails, slugs
	Salads
	Juices Fruits Planarians
	Freshwater shrimp Crabs
	Frogs
	Lizards
	Water contaminated with larvae
Touching, handling	Mollusks, snails, or slugs
Residence or recent travel	Endemic areas

**Table 2 pathogens-12-00624-t002:** Diagnostic criteria for neuroangiostrongyliasis and criteria classification according to the strength of evidence for the etiological diagnosis.

Criteria Category	Criteria	
Minor	a. Exposure history b. Serology (antibody detection) c. Blood eosinophilia	
Major	a. CSF ^1^ eosinophilia b. Headache, other neurological signs or symptoms, and other obvious etiologies ruled out.	
Confirmatory	a. Larvae in tissues, CSF, or eye b. DNA detection ^2^	

^1^ CSF: cerebrospinal fluid. ^2^ Antigen detection in CSF (no test currently available) may be an additional criterion for confirmed diagnosis in the future (see Sears et al. for novel highly sensitive PCR [10]).

**Table 3 pathogens-12-00624-t003:** Classification of diagnostic categories for neuroangiostrongyliasis and recommended actions: follow up and treatment.

Diagnosis Categories	Criteria	Recommended Actions
Suspected		
	Headache, OR other neurological signs/symptoms, OR CSF eosinophilia ^1^ AND Any minor criteria ^2^	Close clinical and laboratory follow up
Probable		
	Headache, OR other neurological signs/symptoms AND CSF eosinophilia ^1^ AND At least two minor criteria ^2^	^3^ Treatment: corticosteroids and albendazole (15 mg/kg, bid, 14 days)
Confirmed		
	Larvae in tissues, CSF, or eye chambers OR DNA detection ^4^	^3^ Treatment: corticosteroids and albendazole (15 mg/kg, bid, 14 days)

If serology is positive ^1^ and CSF eosinophilia ^2^ is higher than 40%, consider NA as highly probable/suspected. ^3^ For details and treatment alternatives, see Sawanyawisuth, K. and Sawanyawisuth [14] and Jacob et al. [26]. ^4^ Antigen detection in CSF (no test currently available) may be an additional criterion for confirmed diagnosis in the future (see Sears et al. for novel highly sensitive PCR [10]).

## Data Availability

All data is presented in the paper.

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
