# Peer review of "Neuroangiostrongyliasis: Updated Provisional Guidelines for Diagnosis and Case Definitions"

_pathogens, 2023, doi:10.3390/pathogens12040624_

Round 1

Reviewer 1 Report

This paper is a welcomed addition to the literature on neuroangiostrongyliasis as it seeks to establish a case definition for a parasite that occurs sporadically in humans with severe clinical consequences.  The absence of well-established databases with well define sera and CSF is a major hindrance to the evaluation of diagnostic assays. 

The paper requires revision by a native English speaker; however, more importantly, the Discussion of the paper lacks focus on the major aims which were guidelines for diagnosis and case definitions.  While the paper went into detail on these subjects they were not adequately discussed.

Author Response

Reviewer 1

This paper is a welcomed addition to the literature on neuroangiostrongyliasis as it seeks to establish a case definition for a parasite that occurs sporadically in humans with severe clinical consequences.  The absence of well-established databases with well define sera and CSF is a major hindrance to the evaluation of diagnostic assays. 

1.1) The paper requires revision by a native English speaker; however, more importantly, the Discussion of the paper lacks focus on the major aims which were guidelines for diagnosis and case definitions.  While the paper went into detail on these subjects they were not adequately discussed.

Response: The manuscript was edited by a native-speaker, professional service (“Write Science Right”).  It was also thoughtfully revised by several native-speaker co-authors.  We thank the reviewer appreciation of the detailed approach about diagnostic criteria and case definitions.  The present report is not intended to go into further details, that have been well addressed by several published reviews.

Reviewer 2 Report

Interesting paper giving guidelines for the management of neuroangiostrongyliasis. It is written by a batch of scientists specialized in the domain and seems of utmost importance for this difficult diagnosis.

Line 74: The disease is also observed in other parts of the world which could be specified here:    Australia, South America, United States, and some islands of the Caribbean (Cuba, Haiti, Dominican Republic, Jamaica, Martinique and Guadeloupe).

For west Indies this paper could be mentioned:  Dard C, Tessier E, Nguyen D, Epelboin L, Harrois D, Swale C, Cabié A, de Meuron K, Miossec C, Desbois-Nogard N. First cases of Angiostrongylus cantonensis infection reported in Martinique, 2002-2017. Parasite. 2020;27:31. doi: 10.1051/parasite/2020032. 

In figure 1:  I do not understand "antibody exposure" it should be "antibody detection"?

Please check by an English American pharmacologists that "15 mg /kg−1

day−1 bid" means "15mg daily in two divided doses"... 

Table 2: are the dots necessary? In the table Suspected, Probable and Confirmed should be in the column "Diagnosis categories" and not in the column "Criteria"

line 219: albendazole regimens could be given here...If you are using such guidelines, it could be interesting to have here short guidelines treatment. Moreover, treament is detailed in figure 1. Same remark for corticosteroids: is prednisolone the recommended drug?

References:

Angiostrogylus cantonensis  should be in italics lines 255, 304, 315, 323, 326, 332

Author Response

Reviewer 2

Interesting paper giving guidelines for the management of neuroangiostrongyliasis. It is written by a batch of scientists specialized in the domain and seems of utmost importance for this difficult diagnosis.

2.1) Line 74: The disease is also observed in other parts of the world which could be specified here:    Australia, South America, United States, and some islands of the Caribbean (Cuba, Haiti, Dominican Republic, Jamaica, Martinique and Guadeloupe).

For west Indies this paper could be mentioned:  Dard C, Tessier E, Nguyen D, Epelboin L, Harrois D, Swale C, Cabié A, de Meuron K, Miossec C, Desbois-Nogard N. First cases of Angiostrongylus cantonensis infection reported in Martinique, 2002-2017. Parasite. 2020;27:31. doi: 10.1051/parasite/2020032.

Response: Please see also the response to item 1.1 (Reviewer 1).  As the main focus of the present manuscript is to review the diagnostic criteria and propose a provisional case definition, but not a full review of so many interesting aspects of cerebral angiostrongyliasis (with many well-written published papers) we refrain to go into details.  We thank reviewer 2 for the suggestion, and we identify the need for collaborative studies to update geographic distribution of angiostrongyliasis active transmission foci, with a especial emphasis on the Caribbean region.

2.2) In figure 1:  I do not understand "antibody exposure" it should be "antibody detection"?

Please check by an English American pharmacologists that "15 mg /kg−1

day−1 bid" means "15mg daily in two divided doses"... 

Response: Agreed. We will accondingly correct the Figure.

2.3) Table 2: are the dots necessary? In the table Suspected, Probable and Confirmed should be in the column "Diagnosis categories" and not in the column "Criteria"

Response: Agreed. The original Word file had no centre alignment.  It was automatically changed when PDF files were built. See also comments and response to reviewer 3 (3.7). We will ask help from the editorial office to optimize formatting.

2.4) Line 219: albendazole regimens could be given here...If you are using such guidelines, it could be interesting to have here short guidelines treatment. Moreover, treament is detailed in figure 1. Same remark for corticosteroids: is prednisolone the recommended drug?

Response: Please see also our response to item 1.1 (reviewer 1).  To date, albendazole regimen is well established (although without clear beneficial role in neuroangiostrongyliasis), as presented in the Table.  We think duplication of this information in the text is not necessary. Corticosteroids are clearly beneficial, and prednisolone has been evaluated in well-designed clinical trials, what does not exclude other steroid drugs.   Especially for treatment details we refer to relevant publications.   We also clearly need collaborative-international clinical trials for better evaluation of treatment regimens.  These provisional guidelines are the starting point for this effort.

2.5) References:  Angiostrogylus cantonensis  should be in italics lines 255, 304, 315, 323, 326, 332

Response: Agreed and done.

Reviewer 3 Report

I read with interest the paper "Neuroangiostrongyliasis: updated provisional guidelines for diagnosis and case definitions". I found some minor issues which I annotated and commented on in the manuscript-pdf-file.

Author Response

Reviewer 3

I read with interest the paper "Neuroangiostrongyliasis: updated provisional guidelines for diagnosis and case definitions". I found some minor issues which I annotated and commented on in the manuscript-pdf-file

3.1) Lines 46-47: Such general statements are problematic. In a global context or in regions with high burden of parasitic diseases the statement may be correct. In e.g. central Europe this statement is clearly wrong, as in this epidemiological setting non-parasitic cases of eosinophilic meningoencephalitis are predominant. I propose a more differentiated phrasing.

Response: phrasing updated in Line 52: “The non-infectious causes for EoM are dominant in many regions, e.g. central Europe.”

3.2) Line 52: I suggest deleting "also" as semantically there is no corresponding context to which it refers ("also" here would only be correct if previously other paratenic hosts would "also" have been mentioned)

Response: “also” was removed, as suggested by reviewer 3.

3.3) Line 68: Add plausible references to substantiate this claim. One may question expansion and rather suggest increasingly available awareness as well as wider availability of diagnostics for the "emergence"/"expansion" of reports from previously unaffected regions.

Response: We agree increased awareness may have a role in the perceived expansion of neuroangiostrongyliasis.  Added phrase (updated Line 72): “This perceived expansion may have also resulted from an increased awareness and availability of diagnostic capability.”

3.4) Lines 70-73: I suggest to remove this overdramatic note, as no travel medicine specialist I know (and I work in the business since one and a half decades) would spend time specifically addressing the risk of angiostrongyliasis... The number of cases in travellers are far too low and there is no indication of a substantial increase in cases over time to reasonably claim "concern". Note on my last comment above: Considering the reduced rate of occurrence in endemic regions the above mentioned "rising concern" regarding travelers is an obvious contradiction.

Response: Suggestion is accepted. We removed “…a recent expansion of transmission areas, and a rising concern for travelers,…”;  updated Line 75: “Despite the potential for severe CNS disease, NA is not a highly prevalent infection worldwide.”

3.5) Line 77: "Mortality" refers to the number of deaths per unit of the general population. The mentioned 80% is "lethality" = number of deaths / over number of sick with a specific disease.

Response: Agreed. “Lethality” replaces “mortality”, updated Line 82.

3.6) Figure: The target of the in-figure reference "5" is missing/unclear. For the in-figure number "2" a legend is missing. Add a legend clarifying these 2 figures.

Response: Agreed. The numbers “5” and “2” are misleading and unnecessary.  They will be removed.

3.7) Tables: No centre alignement in tables please. This format is not supporting that the reader can easily catch the content. Change this format in all tables.

Response: Agreed. The original Word file had no centre alignment.  It was automatically changed when PDF files were built.

3.8) Table 2: I object suggesting the use of "other benzimidazole drugs": the currently only commercially available benizimdazole drug, capable of crossing the blood–brain barrier, around the globe is albendazole. It is also the only benizimidazole studied with relevant data in the context of treating neuroangistrongyliasis. Can the authors explain, which alternative benzimidazole they address and reference the evidence to consider this compound to be an alternative?

Response: Agreed, “…or other benzimidazole drug” was removed.

3.9) Line 185: This sentence suggests that in Hawaii infection arises solely from nonintentional ingestion of infected mollusks... A very doubtful claim, considering that ingestion of vegetables contaminated by mollusk/slug slime containing larvae is very likely more frequent. I understand what the authors want to says, but phrasing it this way is incorrect in my opinion.

Response: Agreed. New phrasing (now Line 196): “For example, in Thailand, intentional food habits favor transmission; while in Hawaii, ingestion of contaminated food or water, is usually nonintentional.”

3.10) Line 198: Just a note: The practical clinical relevance of this laboratory observation is practically nil. The limited time ingested larvae would be  reachable by a non-absorbed antihelminthic in the GI tract is prohibiting its usefulness for individual patients as well as in the context of clusters. The only thinkable us as a post-exposure prophylaxis (PEP) with pyrantel would thus be in a setting where intentional ingestion of wild (non-culture raised) mollusks is practiced. Pretty unrealistic...

I would refrain from mentioning things without clearly putting them in context. The result is that clinicians like me have to deal with patients demanding PEP because they ate in a restaurant some weeks ago in which a patient may have contracted angiostrongiliasis...

Response: Agreed. Rephrased: (updated Line 209) “Post-exposure treatment with pyrantel has been shown to be promising, although its clinical relevance is yet to be demonstrated, especially considering the very short time frame, from exposure to an effective prevention of larvae migration through intestinal mucosa”

3.11) Line 203: Only a note: I highly appreciate this, as we have a substantial number of patients with a positive serology result resulting from the undifferentiated order - mostly sadly initiated by general practitioners - in cases where the pretest probability is close to or practically nil... Explaining the patient the complexity of serology result interpretation is unfortunately consuming a substantial time in our clinic... Thus, the clear downgrading of serology is a very appreciated step which hopefully trickles down to the ones ordering these tests.

Response: thanks for the comments.